# A double-blind, randomized controlled trial to examine the effect of *Moringa oleifera* leaf powder supplementation on the immune status and anthropometric parameters of adult HIV patients on antiretroviral therapy in a resource-limited setting

**Aisha Gambo**[1]*, **Indres Moodley**[1], **Musa Babashani**[2,3], **Tesleem K. Babalola**[1], **Nceba Gqaleni**[4]

**1** Discipline of Public Health Medicine, School of Nursing and Public Health, College of Health Sciences, University of KwaZulu-Natal, Durban, South Africa, **2** Department of Medicine, Bayero University Kano, Kano State, Nigeria, **3** Aminu Kano Teaching Hospital, Kano State, Nigeria, **4** Discipline of Traditional Medicine, School of Nursing and Public Health, College of Health Sciences, University of KwaZulu-Natal, Durban, South Africa

\* gamboaishatu@yahoo.com

## Abstract

### Background

People living with HIV (PLHIV) in resource-limited settings are vulnerable to malnutrition. Nutritional interventions aimed at improving food insecurity and malnutrition, together with antiretroviral therapy (ART), could improve treatment outcomes. In Nigeria, there is a high awareness of the nutraceutical benefits of *Moringa oleifera*. Thus, this study aimed to evaluate the effects of *Moringa oleifera* leaf supplementation on the CD4 counts, viral load and anthropometric of HIV-positive adults on ART.

### Methods

This was a double-blind, randomized study. Two hundred HIV-positive patients were randomly allocated to either the *Moringa Oleifera* group (MOG) given *Moringa oleifera* leaf powder or the control group (COG) given a placebo. Changes in anthropometric parameters [weight; body mass index (BMI)] and CD4 cell counts were measured monthly for six months, while HIV-1 viral loads were measured at baseline and the end of the study for both groups.

### Results

Over the study period, the treatment by time interaction shows a significant difference in CD4 counts by treatment group (p<0.0001). A further estimate of fixed effects showed that the CD4 counts among MOG were 10.33 folds greater than COG over the study period.

**Data Availability Statement:** Data cannot be shared publicly due to the ethical restrictions regarding patient confidentiality imposed by the ethics committee of Aminu Kano Teaching Hospital, Kano state, Nigeria and University of Kwazulu- Natal, Durban, South Africa. Interested and qualified researchers who meet the criteria for access of data can request data access from the: Biomedical Research Ethics Committee of University of Kwazulu-Natal, Durban South Africa. Westville Campus, Govan Mbeki Building. Postal address: Private Bag x54001, Durban 4000. Tel: +27(0) 31 260 2486, Facsimille: +27(0) 31 260 4609. Email: brec@ukzn.ac.za.

**Funding:** This study was funded by the Department of Science and Innovation of South Africa (DST/ CON 0196/2011) and the College of Health Sciences, University of KwaZulu-Natal, Durban, South Africa. The funders had no role in study design, data collection and analysis, decision to publish, or preparation of the manuscript. The authors received no specific funding for this work.

**Competing interests:** The authors have declared that no competing interests exist.

However, the viral load (p = 0.9558) and all the anthropometric parameters (weight; p = 0.5556 and BMI; p = 0.5145) between the two groups were not significantly different over time. All tests were conducted at 95CI.

## Conclusion

This study revealed that *Moringa oleifera* leaf supplementation was associated with increased CD4 cell counts of PLHIV on ART in a resource-limited setting. Programs in low-resource settings, such as Nigeria, should consider nutritional supplementation as part of a comprehensive approach to ensure optimal treatment outcomes in PLHIV.

## Introduction

The HIV and AIDS epidemic is a major pandemic that affects millions of people globally. The UNAIDS Global AIDS Update 2019 reported that 74.9 million people have become infected with HIV since the start of the epidemic, with 32.0 million deaths from AIDS-related illnesses [1]. Nigeria has the second-largest HIV epidemic worldwide [2]. In 2018, 130,000 new infections and 53,000 AIDS-related deaths were recorded. Nigeria alone accounts for more than half of the new infections and deaths from AIDS-related illnesses in the western and eastern Africa region in 2017 [3]. This high mortality is probably attributed to the large population size of Nigeria compared to other countries in the region [3].

Considerable progress has been made in providing global access to antiretroviral therapy (ART), with 23.3 million people accessing therapy worldwide [4]. ART has greatly reduced AIDS-related mortality and morbidity and increased the life expectancy of people living with HIV and AIDS (PLHIV) [5]; however, it has led to other consequences, including malnutrition [6]. PLHIV are vulnerable to malnutrition due to intestinal damage, which causes impaired nutrient absorption and reduced food intake from vomiting and painful swallowing [7]. Furthermore, malnutrition could result from food insecurity and the side effects of ART, such as appetite loss and abdominal pain [7]. The adverse effects of HIV and malnutrition on the immune system are similar in that they both reduce CD4 and CD8 T-lymphocyte numbers [8], which eventually increase susceptibility to opportunistic infections. Opportunistic infections and malnutrition can affect intake, absorption, and metabolism of food, worsen disease progression [7, 9] and increase HIV-related mortality [6].

PLHIV are encouraged to consume healthy diets rich in essential amino acids, unsaturated fats, and micronutrients at the recommended daily allowance (RDA) to achieve an adequate nutritional status vital for health and survival [10]. Unfortunately, several studies reported a poor diet intake with inadequate nutrients among PLHIV in Sub-Saharan Africa, including Nigeria [11, 12]. A study conducted in Nigeria reported significant malnutrition in early HIV infection before ART initiation [13].

Malnutrition is a public health challenge in Nigeria; available data showed that the country has the second-highest burden of stunted children worldwide [14]. Two million children and 7% of women of childbearing age were also reported to suffer from severe acute malnutrition [14].

*Moringa oleifera* Lam (syn. M. ptreygosperma Gaertn.) is a species of the monogeneric family Moringaceae [15, 16]. It has been documented to contain many nutrients and bioactive compounds in literature [17, 18]. The leaves are the part of the plant mostly used and with several nutrients often deficient in malnourished PLHIV. It is a rich source of both macro and

micronutrients and natural antioxidants source [19]. *Moringa oleifera* leaf powder is a novel, cheap, culturally acceptable, efficacious, and regionally produced plant and can reduce the malnutrition burden in Sub-Saharan Africa [19]. Furthermore, in Nigeria, *Moringa oleifera* use is promoted based on the commendation of its nutraceutical benefits, and the Nigerian Federal Government Raw Materials Research and Development Council (RMRDC) has been actively encouraging farming and consumption of *Moringa oleifera* [20]. Monera *et al*. reported the *in vitro* CYP3A4 inhibitory activity of *Moringa oleifera* leaf extracts, suggesting the potential for interaction with antiretroviral drugs. However, the *in vitro* data alone is insufficient to conclude the clinical significance of concomitant administration of *Moringa oleifera* with ART in PLHIV [21]. Moreover, the interaction between tenofovir/lamivudine/efavirenz and *Moringa oleifera* leaf powder has not been reported. No adverse clinical effects have been reported in the literature despite its widespread use and concomitant use by PLHIV.

Therefore, this double-blind, randomized study aimed to evaluate the effects of six months of *Moringa oleifera* leaf supplementation on the CD4 counts, viral load and anthropometric parameters of HIV-positive adults who were on ART in Kano State, Nigeria.

## Methods

### Study location

The study was conducted at the S. S Wali Virology Center at the Aminu Kano Teaching Hospital, Kano State (AKTH), Nigeria. AKTH is a tertiary health institution and referral center that operates a daily HIV clinic (5 days a week). It also serves as a center for clinical evaluation, laboratory tests, HIV counseling and testing (HCT), treatment, and care supported by the Federal Government and the Institute of Human Virology, Nigeria (IHVN) in partnership with its global partners, including the Centers for Disease Control and Prevention (CDC) and the Global Fund to Fight AIDS, Tuberculosis, and Malaria. The center attends to all patients with HIV infection diagnosed within the hospital or referred from outside the health facility.

### Type of study and participants

The study was a double-blind, randomized control trial conducted between December 2017 and November 2018. Registered HIV-infected individuals receiving treatment and care at the S.S Wali Virology Center were invited to participate. Inclusion criteria for the study were: being HIV sero-positive, $\geq$ 18 years old, CD4 counts $\leq$ 500 cells/mm$^3$, ART for at least three months (tenofovir + lamivudine + efavirenz combination), informed consent, and compliance with the study protocol. Exclusion criteria for the study were: known allergy or intolerance to *Moringa oleifera* or placebo (cornstarch powder), pregnancy, CD4 counts > 500 cells/mm$^3$, presence of active opportunistic infection, and intake of micronutrient or natural health product supplements within 30 days of screening. For ease of monitoring, patients who lived outside Kano State, where the study was conducted, were excluded.

### Sample size

The sample size was calculated to ensure detection of medium effect size (Cohen's d = 0.5) [22] or 0.5 standard deviation in mean weight or CD4 by randomized control trial (RCT) arm with 90% power (1-β [type 2 error probability]) and 95% confidence (or 5% α error probability [type 1]), assuming a balanced 1:1 study design. A sample size of 172 patients was calculated, rounded up to 200 to give room for attrition. The sample size was calculated using G*Power version 3.1.9.2 [23].

## Randomization

Block randomization was used to balance the groups throughout the enrollment period. PASS 12.0 software was used to develop the randomization list using Wei's Urn algorithm by an independent statistician who held the randomization code. A random allocation sequence was generated to allocate and assign each patient to either the *Moringa Oleifera* group (MOG) or the control group (COG) when participants fulfilled the inclusion criteria and consented, with 100 patients in each group. All the research team members, including the principal investigator (PI) and the study participants, were blinded to the allocation of patients to the study groups.

## Preparation for study

Fresh *Moringa oleifera* leaves were obtained from Prime Global Agricultural Industries Limited, Kano State, Nigeria. Fresh leaves were identified and authenticated by a botanist at the Department of Biological Science, Bayero University Kano (BUK), Nigeria. Confirmation of the taxonomic identity of the plant was achieved by comparison with voucher specimens kept at the Herbarium of the Department of Biological Sciences, BUK. The leaves were processed by HOMIP Spices and Foods Limited, Kano State, Nigeria. The procedure involved washing and drying the fresh leaves in a clean environment on a net mesh away from direct sun for days until it was completely dried. The dried leaves were cleaned, and the small branches were removed. The dried leaves were ground using a grinder and sieved using a 0.500 mm standard sieve (No. 35 mesh size) [24, 25] to obtain a fine powder. Fine Moringa leaf powder was stored in airtight containers.

The placebo was obtained by coloring cornstarch powder with chlorophyll [26]. It was manufactured and processed at Dala Foods Nigeria Limited, Kano State, Nigeria. Both the *Moringa oleifera* and the placebo were similar in presentation and were identically packaged to be indistinguishable. The interventions were packaged into small (15 g) sachets each. Thirty (30) individual sachets were further packaged in a bigger green-colored plastic bag to be used as supplements taken together with meals for one month. It was sealed, labeled, and stored in a dry place away from heat and humidity. Patients could simply put a sachet in the pocket or bag while going out for their daily activities. The supplements were taken together with meals.

## Intervention

The interventions were provided in 15 g Moringa leaf powder sachets. Thirty sachets were given to the participants to represent one month prescription, and they were directed to divide each sachet into three and use it thrice daily (5 g), adding it into meals [27, 28]. They were asked to maintain their regular diet and not consume *Moringa oleifera* in any form from other sources during the study period.

In Kano State, home visits of participants by research members to ensure adherence was not convenient for fear of stigmatization by family members. Thus, adherence was monitored by biweekly phone calls to the patients and interviewing them during their monthly visits to evaluate compliance.

## Data collection

At the first visit, the research team interviewed the patients to obtain socio-demographic information, patient history, and other relevant information, including dietary information. A trained nurse at the virology clinic and a trained research assistant were responsible for all anthropometric measurements and data collection under the supervision of the PI. Weight was measured to the nearest 0.1 kg using a digital weighing scale (Tanita HD-372, Tanita

Corporation, Tokyo, Japan), with participants wearing light clothing and without shoes. Height was measured to the nearest centimeter using a stadiometer (Seca 217, Seca Gmbh and co. KG., Hamburg, Germany). Body mass index (BMI) was calculated as the weight in kilograms divided by the square of height in meters. Anthropometric parameters were measured at baseline and each monthly visit.

All laboratory evaluations were performed by a trained phlebotomist at the President's Emergency Plan for AIDS Relief (PEPFAR) laboratory of the S. S Wali Virology Center at AKTH. The CD4 count was tested using a Partec flow cytometer (Partec, Munster, Germany). Five (5 ml) venous blood samples were aseptically collected from each study participant. Briefly, equal volumes (20 μL) of CD4 PE antibody and ethylene diamine tetraacetic acid blood were mixed and incubated for 15 min, and 800 μL of CD4 buffer was added before reading in the cell counter [29]. The viral load was quantified by polymerase chain reaction (PCR). Ten (10 ml) of venous blood samples were aseptically collected from each study participant. The COBAS AmpliPrep/COBAS TaqMan HIV-1 Test version 2.0, manuals (Roche Diagnostics GmbH) was used as the standard operating procedure [30, 31]. The CD4 test was conducted at baseline and each subsequent monthly visit for each study participant, while the viral load test was conducted twice, at baseline and after the sixth month.

## Study outcomes

The outcomes assessed were changes in immune status (CD4 cell count and viral load) and changes in anthropometric parameters (weight and body mass index [BMI]) and from baseline to the sixth month.

## Data analysis

The data input was done in Microsoft excel and exported into SPSS and SAS statistical software for analysis. Findings from the analysis were reported in frequency tables, charts and descriptive analysis was done to estimate mean and standard deviation. Normality test for the data was conducted using Kolmogorov-Simonov and Shapiro-Wilk tests. Data which were not normally distributed were transformed through Box-Cox transformation. Independent t-test was used to determine the significance of mean difference in immunological and anthropometric parameters between the two groups at each stage of the experiment. To further confirm variability between the two groups, a repeated measure linear mixed effect model analysis was deployed to determine the difference in immunological and anthropometrics outcomes between the treatment groups overtime. An exploratory analysis was done to evaluate the influence of socio-demographic characteristics on the immunological and anthropometrics outcomes of the treatment groups over the study period. All statistical tests were carried out at 95% Confidence Interval.

## Ethical considerations

This study was reviewed and approved by the ethics committee of Aminu Kano Teaching Hospital (AKTH) Kano, Nigeria (reference number NHREC/21/08/2008/AKTH/EC/2012), and the Biomedical Research Ethics Committee of the University of Kwazulu-Natal Durban, South Africa (reference number BFC294/16). The study was registered with the Pan African Clinical Trial Registry (identification number PACTR201811722056449). The study complied with the principles outlined in the Declaration of Helsinki [32]. All participants provided oral or written informed consent before enrolling them in the study. The procedures of the study, together with the aims, were explained to the participants. Participants were also informed of their right to withdraw from the study at any time.

## Results

### Participants flow

Fig 1 shows the flow chart of participants in the study. Four hundred and ten patients were screened and assessed for eligibility. Two hundred and ten patients were excluded (204 did not meet the inclusion criteria for the study, and 6 refused to participate). Two hundred patients were randomized into two groups. One hundred patients were randomly selected and allocated to the group receiving MOG, and 100 patients were randomly assigned to the group receiving COG. In the MOG, 8 patients were lost to follow-up, and 3 discontinued the intervention. In the COG, 9 patients were lost to follow-up, and 3 discontinued the intervention. Overall, 177 patients (89 and 88 in the MOG and COG, respectively) completed the 6-month study.

### Nutritional contents of *Moringa oleifera* leaves powder

The nutritional content of a 100 g *Moringa oleifera* leaf powder (Nigerian ecotype) was analyzed using a South African National Accreditation System (SANAS) [33] accredited laboratory ASPIRATA Food and Beverage Laboratory [34]. Each 100 g contained an average of 28 g protein, 3.9 g total fat content (total saturated, monounsaturated, and polyunsaturated fatty acids), and 22 g carbohydrate. It contained 1791.82 mg calcium; 4879.26 mg potassium; 24 mg sodium; 2.88 mg Zinc and 37.78 mg iron (S1 & S2 Files).

### Characteristics of participants at study inception

Table 1 shows the socio-demographic characteristics of the study participants at baseline. Participants in the MOG's socio-demographic, socioeconomic, nutritional status, and immunological characteristics were similar to those in the COG at baseline. Females were predominant in both groups [MOG = 70 (78.7%); COG = 67 (76.1%)]. The majority were between 30 to 39 years of age in both groups [MOG = 37 (41.6%); COG = 36 (40.9%)]. The majority were married [MOG = 42 (47.2%); COG = 38 (43.2%)]. Islam was the predominant religion of participants in both groups [MOG = 64 (71.9%); COG = 66 (75%)] with more than half of participants belonging to the Hausa/Fulani ethnicity [MOG = 55 (61.8%); COG = 47 (53.4%)]. A few of the participants were without any form of education in either group [MOG = 15 (16.9%); COG = 16 (18.2%)]. The majority of the participants in both groups earned a monthly income below the minimum wage of ₦30,000 ($78.23) [MOG = 67 (75.3%); COG = 66 (75%)].

The baseline anthropometric and immunological characteristics of the study participants in both study groups showed a similar trend. The means of weight (kg) for both groups were [MOG = 63.8 (± 14.8); COG = 61.9 (±12.5)]. The mean BMIs for MOG and COG was 24.84 (± 4.76) and 23.75 (± 3.82), respectively. More than half of the patients had BMI within the normal range of 18.5–24.9 in both groups [MOG = (51.7%); COG = (58%)] while a significant number were overweight with BMI values of 25.0–29.9 [MOG = (30.3%); COG = (31.8%)] for both study groups (Table 2).

At baseline, the mean CD4 cell counts were statistically similar for both MOG and COG with values of 341.78 (± 106.06) and 352.34 (± 125.99) cells/μL, respectively (Table 2). The majority of the patients in both groups had an undetected viral load at baseline (MOG = 76.4% and COG = 71.6%) (Table 2).

A test of normality for the dependent variables was carried out using the Shapiro-Wilk and Kolmogorov-Smirnov tests. The data that were not normally distributed were transformed through Box-Cox transformation. The CD4 count was transformed through a lambda value of 0.5, weight by lambda value -0.1 and BMI by lambda value -0.2.

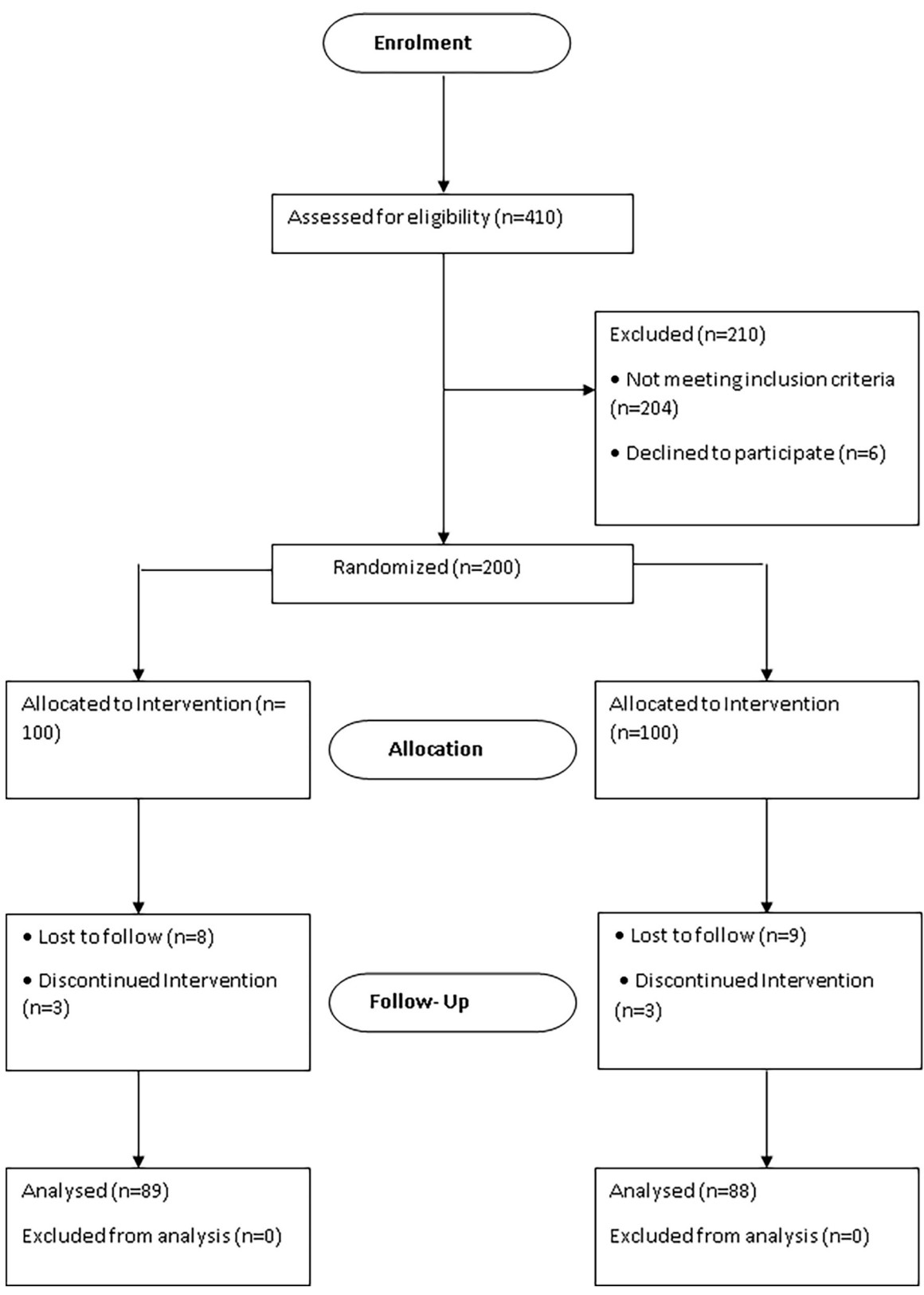

**Fig 1. Flow chart of participants.**

Table 1. Socio-demographic characteristics of participants.

| Variables | MOG (%) (N = 89) | COG (%) (N = 88) | P-value |
|---|---|---|---|
| **Gender** | | | |
| Males | 19 (21.3) | 21 (23.9) | 0.689 |
| Female | 70 (78.7) | 67 (76.1) | |
| **Age (years)** | | | |
| < 20 | 3 (3.4) | 1 (1.1) | 0.737 |
| 20–29 | 24 (27.0) | 21 (23.9) | |
| 30–39 | 37 (41.6) | 36 (40.9) | |
| 40–49 | 20 (22.5) | 22 (25.0) | |
| 50–60 | 5 (5.6) | 8 (9.1) | |
| **Marital Status** | | | |
| Married | 42 (47.2) | 38 (43.2) | 0.838 |
| Single | 12 (13.5) | 10 (11.4) | |
| Divorced | 19 (21.3) | 20 (22.7) | |
| Widowed | 16 (18.0) | 20 (22.7) | |
| **Religion** | | | |
| Islam | 64 (71.9) | 66 (75.0) | 0.642 |
| Christianity | 25 (28.1) | 22 (25.0) | |
| **Ethnicity** | | | |
| Hausa/Fulani | 55 (61.8) | 47 (53.4) | 0.511 |
| Yoruba | 13 (14.6) | 15 (17.0) | |
| Igbo | 9 (10.1) | 15 (17.0) | |
| Others | 12 (13.5) | 11 (12.5) | |
| **Educational Level** | | | |
| Primary | 14 (15.7) | 12 (13.6) | 0.971 |
| Secondary | 27 (30.3) | 24 (27.3) | |
| Tertiary | 20 (22.5) | 21 (23.9) | |
| Quranic | 13 (14.6) | 15 (17.0) | |
| None | 15 (16.9) | 16 (18.2) | |
| **Occupation** | | | |
| Entrepreneur | 15 (16.9) | 10 (11.4) | 0.840 |
| Trader | 23 (25.8) | 25 (28.4) | |
| Civil Servant | 15 (16.9) | 17 (19.3) | |
| Artisan | 19 (21.3) | 17 (19.3) | |
| Unemployed | 17 (19.1) | 19 (21.6) | |
| **Family Size** | | | |
| 2–5 | 38 (42.7) | 32 (36.4) | 0.557 |
| 6–10 | 26 (29.2) | 25 (28.4) | |
| >10 | 25 (28.1) | 31 (35.2) | |
| **Monthly Income(₦)** | | | |
| Not Indicated | 11 (12.4) | 6 (6.8) | 0.672 |
| < 30,000 | 67 (75.3) | 66 (75.0) | |
| 30,001–60,000 | 6 (6.7) | 10 (11.4) | |
| 60,001–90,000 | 1 (1.1) | 1 (1.1) | |
| 90,001–120,000 | 3 (3.4) | 2 (2.3) | |
| >120,000 | 1 (1.1) | 3 (3.4) | |

**Table 2. Description of baseline anthropometric and immunological parameters between the two groups.**

| Parameters | Baseline | | P-value |
|---|---|---|---|
| | MOG (n = 89) | COG (n = 88) | |
| | Freq. (%) | Freq. (%) | |
| **Anthropometric** | | | |
| **Weight** (Kg) | | | **0.361** |
| Mean (±SD) | 63.8 (±14.8) | 61.9 (±12.5) | |
| **BMI** (Kg/m$^2$) | | | **0.093** |
| Underweight (<18.5) | 5 (5.6) | 5 (5.7) | |
| Normal (18.5–24.9) | 46 (51.7) | 51 (58.0) | |
| Overweight (25.0–29.9) | 27 (30.3) | 28 (31.8) | |
| Obese (> 30.0) | 11 (12.4) | 4 (4.5) | |
| Mean (±SD) | 24.84 (±4.8) | 23.75 (±3.8) | |
| **Immunological Parameters** | | | |
| **CD4 Counts** (Cells/µl) | | | **0.547** |
| < 350 | 46 (51.7) | 38 (43.2) | |
| ≥ 350 | 43 (48.3) | 50 (56.8) | |
| Mean (±SD) | 341.8 (±106.1) | 352.34 (±126.0) | |
| **Viral load** (RNA copies/ml) | | | |
| <1000 (Undetected) | 68 | 63 | **0.497** |
| ≥1000(Detected) | 21 | 25 | |

## Effect of nutritional supplement intervention on immunological and anthropometric parameters

Table 3 shows the results of independent samples test for the difference in immunological parameters anthropometric between the MOG and COG. The mean CD4 count between the two groups was not significantly different throughout the period of measurement except at the 6th month. From baseline to the 6th month, there was no significant (P>0.05) difference in all the anthropometric parameters [weight; BMI] between the MOG and COG (Table 3).

In addition to the bivariate analysis test above, a linear mixed-effect model was used to examine the differences in anthropometric and immunological parameters between the MOG and COG. Table 4 shows the linear mixed effect model results showing the differences in the CD4 counts, viral load, weight and BMI between the two groups over the study period. An unstructured correlation matrix was assumed for the model analysis. For CD4 counts, the treatment by time interaction shows a significant difference in CD4 counts by treatment group over time (p<0.0001). A further estimate of fixed effects showed that the CD4 counts among MOG were 10.33 folds greater than COG over the study period. On the other hand, viral load (p = 0.9558) and the anthropometric parameters (BMI; p = 0.5145 and weight; p = 0.5556) between the two groups were not significantly different over time (Table 4).

Fig 2 shows the chart depicting CD4 cell count mean measurements by treatment group over the study period. Over the six months study period, there was significant increase in mean CD4 cell counts for MOG while the mean CD4 cell counts for the COG was relatively constant.

An exploratory analysis of the influence of socio-demographic characteristics on the changes in immunological and anthropometric parameters by treatment groups over time was computed. The analysis of the impact of socio-demographic characteristics on the changes in CD4 counts by treatment group over time showed that ethnicity (p = 0.0491) and family size

**Table 3. Bivariate analysis showing the differences in anthropometric and immunological parameters between the two study groups.**

| Parameters | Period | MOG (n = 89) | 95% Confidence Interval | | COG (n = 88) | 95% Confidence Interval | | F | P-value |
|---|---|---|---|---|---|---|---|---|---|
| | | Mean (SD) | Lower | Upper | Mean (SD) | Lower | Upper | | |
| **CD4 Counts** | Baseline | 341.78 (106.06) | 319.43 | 364.12 | 352.34 (125.99) | 325.64 | 379.03 | 4.88 | 0.55 |
| | 1st month | 363.06 (127.91) | 336.11 | 390.00 | 361.14 (130.28) | 333.53 | 388.74 | 0.27 | 0.92 |
| | 2nd month | 373.74 (130.79) | 346.19 | 401.29 | 366.40 (144.47) | 335.78 | 397.01 | 0.50 | 0.72 |
| | 3rd month | 387.29 (134.61) | 358.94 | 415.65 | 367.51 (142.24) | 337.37 | 397.65 | 0.98 | 0.34 |
| | 4th month | 401.51 (138.50) | 372.33 | 430.68 | 368.78 (150.89) | 336.81 | 400.76 | 0.24 | 0.14 |
| | 5th month | 414.79 (144.02) | 384.45 | 445.13 | 375.26 (152.18) | 343.01 | 407.51 | 0.08 | 0.08 |
| | 6th month | 425.75 (153.76) | 393.36 | 458.14 | 373.44 (157.31) | 340.11 | 406.77 | 0.02 | 0.03* |
| **Weight** | Baseline | 63.83 (14.77) | 60.64 | 66.73 | 61.94 (12.54) | 59.45 | 64.82 | 1.62 | 0.36 |
| | 1st month | 63.88 (14.89) | 60.80 | 66.89 | 62.03 (12.92) | 59.41 | 64.88 | 1.58 | 0.38 |
| | 2nd month | 64.26 (14.76) | 61.13 | 67.17 | 62.44 (13.26) | 59.68 | 65.37 | 0.75 | 0.39 |
| | 3rd month | 64.31 (14.93) | 61.16 | 67.33 | 62.55 (13.36) | 59.83 | 65.49 | 0.96 | 0.41 |
| | 4th month | 64.47 (14.93) | 61.29 | 67.44 | 62.73 (13.37) | 60.04 | 65.65 | 0.92 | 0.41 |
| | 5th month | 64.73 (15.00) | 61.59 | 67.76 | 62.99 (13.38) | 60.27 | 65.91 | 1.18 | 0.42 |
| | 6th month | 64.71 (15.07) | 61.54 | 67.82 | 63.16 (13.49) | 60.48 | 66.19 | 1.09 | 0.47 |
| **BMI** | Baseline | 24.84 (4.76) | 23.84 | 25.85 | 23.75 (3.82) | 22.94 | 24.56 | 3.52 | 0.09 |
| | 1st month | 24.86 (4.84) | 23.84 | 25.88 | 23.78 (3.93) | 22.94 | 24.61 | 3.25 | 0.10 |
| | 2nd month | 24.99 (4.82) | 23.98 | 26.01 | 23.92 (4.02) | 23.08 | 24.78 | 2.06 | 0.11 |
| | 3rd month | 24.99 (4.88) | 23.97 | 26.02 | 23.96 (4.04) | 23.11 | 24.82 | 1.76 | 0.13 |
| | 4th month | 25.06 (4.87) | 24.03 | 26.08 | 24.04 (4.10) | 23.17 | 24.91 | 1.45 | 0.14 |
| | 5th month | 25.16 (4.93) | 24.12 | 26.20 | 24.14 (4.08) | 23.27 | 25.00 | 1.78 | 0.14 |
| | 6th month | 25.16 (4.93) | 24.12 | 26.20 | 24.19 (4.09) | 23.33 | 25.06 | 2.40 | 0.16 |

* Statistically significant difference between two groups.

**Table 4. Linear mixed effects model framework showing the differences in immunological and anthropometric parameters between the treatment groups overtime.**

| | Estimates of Fixed Effects[a] | | | | | | |
|---|---|---|---|---|---|---|---|
| | Parameter | Estimate | Std. Error | t | Sig. | 95% Confidence Interval | |
| | | | | | | Lower Bound | Upper Bound |
| | Intercept | 356.35 | 13.10 | 27.20 | 0.0001 | 354.29 | 376.09 |
| **CD Counts** | MOG | 10.33 | 2.65 | 3.89 | 0.0001* | 5.12 | 15.54 |
| | COG | 0[b] | 0 | | | | |
| | Intercept | -1.07 | 0.31 | -3.47 | 0.0007 | -2.02 | -0.99 |
| **Viral load** | MOG | -0.005 | 0.09 | -0.06 | 0.9558 | -0.19 | 0.18 |
| | COG | 0[b] | 0 | . | . | . | . |
| | Intercept | 61.92 | 1.47 | 42.02 | 0.0001 | 61.43 | 63.65 |
| **Weight** | MOG | -0.05 | 0.08 | -0.59 | 0.5556 | -0.20 | -0.11 |
| | COG | 0[b] | 0 | . | . | . | . |
| | Intercept | 23.73 | 0.47 | 50.98 | 0.0001 | 23.61 | 24.32 |
| **BMI** | MOG | -0.02 | 0.03 | -0.65 | 0.5145 | -0.08 | -0.04 |
| | COG | 0[b] | 0 | . | . | . | . |

* = statistically significant.

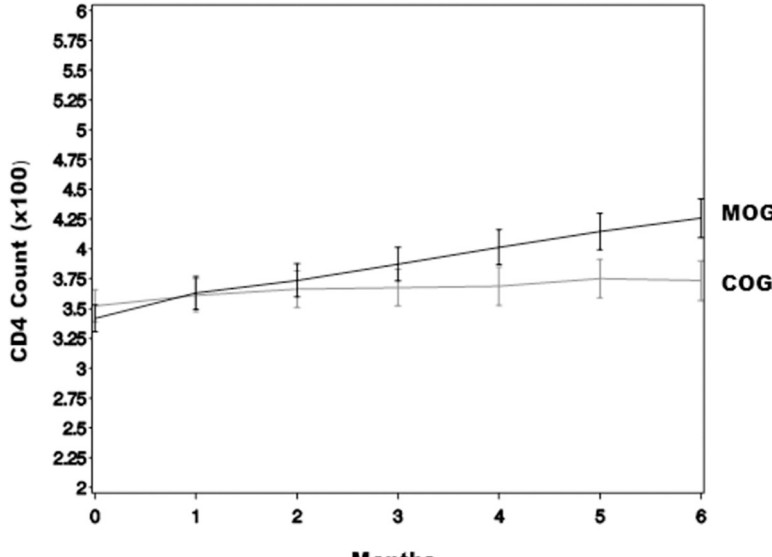

**Fig 2. Chart depicting CD4 cell count mean measurements by treatment group over the study period.**

(p = 0.0483) had a significant influence. However, the changes in CD4 counts by treatment group remained significantly different over time (p = 0.0001). None of the socio-demographic characteristics explored significantly influenced the viral load, BMI, and weight overtime. The changes in these parameters between the treatment groups over time were not significant after controlling for socio-demographic characteristics.

## Discussion

This study examined the effect of the sixth month's consumption of *Moringa oleifera* leaf powder supplement on the immunological profile (CD4 cell count and viral load) and anthropometric parameters (weight and BMI) of PLHIV that are on ART in Kano State, Nigeria. We studied 200 patients randomly divided into MOG (100 patients) and COG (100 patients). Our sample size was larger than that reported by Tshingani *et al*. (60 patients) in the Democratic Republic of Congo and Ogbuagu *et al*. (40 patients) conducted in Anambra State, Southeast Nigeria [10, 35].

Over the study period, a significant increase was observed in CD4 cell counts of the MOG participants than the COG using the linear mixed effect model. There was no significant difference in viral load between the two study groups. This improvement observed in the CD4 counts in the MOG was influenced by some socio-demographic characteristics of the study participants that include ethnicity and family size. However, the improvement was still detected regardless of their influence. This result suggests that *Moringa oleifera* leaf powder supplementation and ART effectively improved the CD4 cell counts of the study participants.

Conversely, the *Moringa oleifera* leaf powder supplementation intervention was not effective in improving the weight and BMI of the patients when compared to the COG over the study period. Furthermore, none of the socio-demographic characteristics explored was observed to significantly influence any of the anthropometric parameters overtime.

*Moringa oleifera* nutritional constituents and ART effect could be responsible for the increased CD4 cell counts in the MOG. *Moringa oleifera* leaves (Nigerian ecotype) analysis shows that they are rich sources of vitamins and micro-and macronutrients. Additionally, it contains minerals and trace elements reported to have multiple curative properties, improve

the immune system, and act as strong antioxidants [17–19]. Furthermore, the dried leaves of *Moringa oleifera* have been documented as a good source of polyphenol compounds, such as flavonoids. Flavonoid consumption has been reported to offer body protection against chronic diseases associated with oxidative stress [18].

In addition, ART has proven effective in reducing morbidity and mortality related to HIV infection by reducing HIV viral load. This improved CD4 level is associated with a reduced number of opportunistic infections [36]. Our results agree with a similar study from Eastern Nigeria by Ogbuagu *et al.*. A significant increase in CD4 counts was observed in PLHIV receiving ART and supplemented their local meals with *Moringa oleifera* leaf powder for two months. Local meals were prepared using palm oil. The limited information offered by the study article prevented an in-depth analysis of the results. However, the study demonstrated the potential of *Moringa oleifera* of improving the CD4 count of PLHIV within short period of supplementing it with regular meals prepared using local food items [35]. Palm oil has been documented to contain palmitic acid which is a saturated fatty acid with health benefits [37].

The lack of significant change in BMI observed in MOG in our study could be due to the fact that few participants (n = 5) were underweight with BMI<18.5 kg/m$^2$ at study inception. This class of people would probably have benefitted more in terms of improvement in BMI from the *Moringa oleifera* leaves supplementation due to the vast amounts of nutrients constituted.

Contrary to our results, Tshingani *et al*. did not report a significant difference in CD4 lymphocyte counts after six months of *Moringa oleifera* leaf powder supplementation between their study groups. This could be due to the small sample size. In addition, the presentation of their intervention in bags of 100 g could have reduced adherence, which resulted in a lack of significant increase in CD4 cell counts [10].

This study's outcomes can be attributed to factors related to the study design. This includes being a double-blinded randomized trial, a larger sample size, and the presentation of the intervention in individual sachets representing daily dose. In addition, the involvement of the virology clinic 'support group' members improved patient monitoring and adherence to the study protocol. Nevertheless, future studies using a more diverse population of PLHIV are recommended.

The *Moringa oleifera* leaf intervention was not effective in decreasing the viral load of HIV-infected individuals accessing ART at the S.S Wali Virology Center. This could be attributed to some of the challenges encountered. The study protocol did not follow the standard operating procedure (SOP) of viral load monitoring conducted at the S. S Wali Virology Center. Viral load was monitored yearly for patients without any medical problems, whereas the study protocol was designed to have a viral load test conducted twice, at baseline and after six months of receiving the intervention. The study plan overcame this challenge. The use of viral load alone without CD4 cell counts to monitor treatment outcomes remains a challenge in resource-limited settings.

Further challenges encountered during the study also include participants' reluctance to keep monthly appointments to the clinic for the study. This is because at the S. S. Wali virology center, ART drugs were dispensed to last for two to three months for patients with stable medical conditions. This challenge was alleviated with the bi-weekly telephone calls performed to monitor the study participants and remind them of their hospital appointments. The stipend given for transport fare after each monthly hospital visit assisted the study participants in keeping their appointments. This is because of the high poverty level in resource-limited settings such as Kano State and the poor socioeconomic status of the study participants. No other incentives or gifts were provided to participants.

Some limitations of this study must be noted. The distinguishable taste of *Moringa oleifera* could be a source of bias. Use of *Moringa oleifera* in capsules could forestall this limitation in future studies. In addition, the inclusion of patients who were only on one ART regimen (tenofovir + lamivudine + efavirenz drug regimen) limits the generalizability of our study findings. Lastly, the short duration of the study and compliance, which was monitored by the self-reporting of the study participants, are further limitations of the study.

## Conclusion

This study revealed an association between *Moringa oleifera* leaf nutritional supplementation consumption and increased CD4 cell counts among PLHIV on ART in a limited resource setting. Programs in low-resource settings, such as Nigeria, should consider nutritional supplementation as part of a comprehensive approach to ensure optimal treatment outcomes in PLHIV.

## Supporting information

**S1 Checklist. CONSORT checklist.**
(DOC)

**S1 File. Certificate of analysis Moringa oleifera powder.**
(PDF)

**S2 File. Certificate of analysis Moringa oleifera powder_Minerals.**
(PDF)

**S3 File. Detailed statistical analysis of data.**
(DOCX)

**S4 File. Study protocol.**
(DOCX)

## Acknowledgments

The authors wish to acknowledge all the patients who participated in this study. We also thank the S. S. Wali Virology Center staff, particularly Nurse Murjanatu Abdulmumin who assisted in the study coordination. Support group members who are also PLHIV and are currently accessing treatment and care at AKTH supported the conduct of the study in ensuring adherence to the research protocol by the participants. We wish to acknowledge and convey our sincere appreciation to the management and staff of Dala Foods Nigeria Limited, Kano State, who assisted in producing the interventions. Lastly, we wish to acknowledge Marothi Peter Letsoalo of the Centre for the Aids Programme of Research in South Africa (CAPRISA) who assisted with the statistical data analysis.

## Author Contributions

**Conceptualization:** Aisha Gambo, Indres Moodley.

**Data curation:** Aisha Gambo, Musa Babashani.

**Formal analysis:** Tesleem K. Babalola.

**Funding acquisition:** Aisha Gambo, Indres Moodley.

**Investigation:** Indres Moodley.

**Methodology:** Aisha Gambo, Indres Moodley.

**Project administration:** Aisha Gambo, Indres Moodley, Musa Babashani.

**Resources:** Aisha Gambo, Indres Moodley.

**Supervision:** Indres Moodley, Musa Babashani.

**Validation:** Tesleem K. Babalola.

**Writing – original draft:** Aisha Gambo.

**Writing – review & editing:** Aisha Gambo, Indres Moodley, Musa Babashani, Tesleem K. Babalola, Nceba Gqaleni.

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
