## [Decision Letter · Decision Letter 0]

18 Jan 2021

PONE-D-20-30776

A double-blind randomized control trial to examine the effect of Moringa Oleifera
leaf powder supplementation on the anthropometric and immune status of adult HIV
patients on antiretroviral therapy in a resource- limited setting

PLOS ONE

Dear Dr. Gambo,

Thank you for submitting your manuscript to PLOS ONE. After careful consideration, we
feel that it has merit but does not fully meet PLOS ONE’s publication criteria as it
currently stands. Therefore, we invite you to submit a revised version of the
manuscript that addresses the points raised during the review process.

The manuscript has been evaluated by three reviewers, and their comments are
available below. You will see the reviewers have commented on the importance of your
work. However, the reviewers have also raised critical concerns and the manuscript
will need significant revision before it can be considered for publication – you
should anticipate that the reviewers will be re-invited to assess the revised
manuscript, so please ensure that your revision is thorough. I have outlined some of
the key concerns noted by the reviewers below, but you should respond all concerns
mentioned by the reviewers in your response-to-reviewers document. 

The key concerns noted by the reviewers relate to the study intervention and
analysis. Specifically, the reviewers have suggested alternate approaches to the
statistical analysis and presentation of results. Additionally, Reviewer 2 noted
that the distinguishing taste of MOG may be considered a source of potential bias in
this study. Finally, Reviewer 3 noted that inclusion was limited to participants on
one ARV regimen, which limits the generalizability of the findings. These issues
have limitations for the interpretation of the results and should be explored.

Please submit your revised manuscript by Mar 02 2021 11:59PM. If you will need more
time than this to complete your revisions, please reply to this message or contact
the journal office at plosone@plos.org. When
you're ready to submit your revision, log on to https://www.editorialmanager.com/pone/ and select the 'Submissions
Needing Revision' folder to locate your manuscript file.

If you would like to make changes to your financial disclosure, please include your
updated statement in your cover letter. Guidelines for resubmitting your figure
files are available below the reviewer comments at the end of this letter.

We look forward to receiving your revised manuscript.

Kind regards,

Danielle Poole

Staff Editor

PLOS ONE

Journal Requirements:

Reviewers' comments:

Reviewer's Responses to Questions

**Comments to the Author**

1. Is the manuscript technically sound, and do the data support the conclusions?

Reviewer #1: Partly

Reviewer #2: No

Reviewer #3: Partly

2. Has the statistical analysis been performed
appropriately and rigorously? 

Reviewer #1: No

Reviewer #2: Yes

Reviewer #3: No

3. Have the authors made all data underlying the
findings in their manuscript fully available?

Reviewer #1: No

Reviewer #2: No

Reviewer #3: No

4. Is the manuscript presented in an intelligible
fashion and written in standard English?

Reviewer #1: Yes

Reviewer #2: Yes

Reviewer #3: No

5. Review Comments to the Author

Reviewer #1: *** General comments: ***

The manuscript presents interesting data from what appears to have been a well-run
study.

However, the statistical analysis is quite idiosyncratic and fails to control for
multiple comparisons. It should be redone.

In addition, the manuscript suffers from a variety of English syntactic and usage
errors. These should be corrected. There are relatively few in the results section,
they are more frequent in the discussion section, and they are numerous in the
remaining sections. There are too many to list.

*** Specific comments: ***

1) The protocol is quite weak in its description of the statistical methods to be
used for the data, as well as the hypotheses of interest and the specific tests or
contrasts that were to be employed. This weakness appears to have carried over to
the actual statistical analysis. The study design is fairly straightforward, so it
is unclear why a better framework for evaluation could not be specified.

Having said that, the most important test specified in the protocol is the t-test of
the two treatments at the last time point, so this should be presented in any
case.

2) The current layout of the statistical analysis is quite idiosyncratic and
simplistic. The following approach or equivalent should be used instead:

Test baseline differences via the t-test as already performed. This is analogous to a
two-factor analysis of variance (ANOVA) with one fixed effect --- that is,
treatment.

Assess the differences between groups in changes over time by using a linear mixed
effect model framework. The fixed effects are treatment, time, and the treatment by
time interaction. Include a correlation structure to account for the dependence
structure caused by observing the same patient over time. Do not use a compound
symmetry structure (equivalent to "classical" repeated measures) --- this is an
unrealistic assumption here. Instead, either use an autoregressive correlation
structure such as AR(1) (since time points are equally spaced) or use a completely
unstructured correlation matrix.

The test of the interaction of treatment by time provides the answer to the question
of whether the response profile over time differs between treatments. If there is no
interaction effect, that is not necessarily the end of the road, but it is arguably
of primary interest.

Next, perform post hoc comparisons using an adjustment for multiple comparisons such
as the Tukey-Kramer adjustment. It is up to the authors, they can perform an
analysis of either simple effects (comparison between treatments at each time point
or comparison among time points within each treatment) or an analysis of all
pairwise comparisons. Note that the test of the interaction effect is not
technically required to be statistically significant for these tests to be
performed, though this is the usual procedure.

All linear models should be assessed for fit using graphical analysis of the
residuals to check for symmetry, homoscedasticity, and independence. The
Shapiro-Wilk or Kolmogorov-Smirnov test can be used, but is probably of less
importance.

If transformations are required, please report all transformations attempted. Even if
fit is marginal on the original scale, if the results are qualitatively similar to a
fit on a transformed scale then it may be best to report analysis on the original
scale for interpretation.

For completeness's sake, since the protocol specifies it, also include the t-test of
the last endpoints. However, this is probably less powerful than the result that
will be obtained from the linear mixed effects model approach.

3) The evaluation of socio-demographic characteristics (Line 270 forward) should be
redone. This should be done in the framework described above, as an exploratory
analysis. These covariates can be entered as main effects in linear mixed effects
model described above. This will yield answers to the question of whether the
treatment effect remains if the effect of the covariate is statistically corrected,
which is probably the question that the authors have in mind.

4) In some cases, it is difficult to understand what exactly is being presented. For
example, what does Table 5 present? What are the "differences" here? Each table and
figure should have a caption that clearly describes its contents.

5) Figure 2 is really terrible and should be replaced with a standard figure that
shows the behavior of the expected marginal means over time for each group.

6) Lines 317-318: This conclusion seems far too strong for this paper. Has it really
been proved to this degree?

7) Lines 338-344: The authors do not disclose whether similar claims were made by COG
patients. This paragraph seems really close to misinformation as it is currently
worded. If patients from both treatments reported these claims, then how can they be
attributed to Moringa oleifera? All such claims in the discussion need to be backed
up by data analysis in the results. Where is the presentation of these results?

8) The methods section lacks critical details. As an example, on Line 175 the authors
state that viral loads were measured "by the Laboratory scientist using Real-time
PCR machine" (sic). Which machine? What was the protocol? Which laboratory? The same
issue pertains to the CD4 counts.

9) Lines 403-405 overstates the results.

10) The conclusion and recommendation in Lines 417-121 do not seem to be fully
supported by the manuscript.

Reviewer #2: 1. Moringa leaf powder packaged in sachets as used in the MOG
potentially had a taste distinguishable from COG. This most likely violated the
blinding process especially in a community where there was a “high awareness” level
of Moringa’s usefulness as a nutraceutical. This should be considered as significant
source of bias enough to dispute the conclusion of a causal relationship between the
MOG intervention and the observed increase in CD4 count. A more firm conclusion
could have been justified if the powder had been packaged in gelatin capsules which
could mask the taste and also ensure more accurate and consistent dosing of the
product.

2. While literature references were made, the claims that Moringa leaf samples
contained essential and non-essential amino acids, wide range of vitamins, minerals,
carotenoids, polyphenols, phenolic acids, flavonoids, alkaloids, glucosinolates,
isothiocyanates, tannins and saponins were unsubstantiated in this particular study.
Perhaps a description of whether the manufacturer Prime Global Agricultural
Industries Limited routinely analyzed the samples for the claimed ingredients would
have been useful.

3. Addition of the discussion of the potential of interactions between Moringa and
antiretroviral therapy, particularly with efavirenz via alteration of liver
metabolism would enrich the manuscript.

4. It is more scientifically acceptable to use the terms such as “study participants”
or “HIV infected individuals” instead of “HIV patients”.

5. Sentences on line 174 and line 176 need reconstruction

6. There may still be more work before a causal relationship can be claimed between
the consumption of Moringa leaf powder and observation of increased CD4 count.
Review recommends that the conclusion should limited to an association between the
intervention and increased CD4 count, siting the observed shortcomings of the study
design.

Reviewer #3: The manuscript reports findings from a randomised controlled double
blinded trial assessing the effect of Moringa oleifera (MO) leaf powder
supplementation on weight, BMI, MUAC, CD4 count and viral load. This is an important
study given that MO is widely consumed and is an important herb among people living
with HIV. Based on the data, the authors conclude that only the CD4 cell counts
improve significantly (p=0.03) after 6 months concurrent MO leaf supplementation and
ART. The manuscript however has some technical issues in some aspects of the trial
design and statistical analysis.

p11 Selection criteria

L105 - Inclusion of patients on only one regimen limits generalisability of the
effect on CD4 counts to patients on other regimen. Why was this necessary?

L113 - Typically, randomisation follows screening and indicates a participant is
enrolled into the study. However, under exclusion criteria authors state that
patients taking micronutrient or natural health product supplements within 30 days
of randomisation were excluded. Please clarify.

p12 Sample size

L122 - In the abstract the authors talk about significance of the mean increase of
CD4 count from baseline at the 99% CI which is inconsistent with sample size
justification based on a 5% margin of error, i.e. 95% CI.

p12 - 13 Intervention

L135 - How were the MO leaves positively identified? Was a botanist consulted?

L156 - What seive size was used to prepare the intervention? 5g weight seems too
heavy for 1 teaspoon of MO leaf powder.

L158 - Specify that MO from other sources is what was disallowed.

p14

L174, L177 - specify the type and volume of blood samples collected and a brief
explanation or reference to the details of the laboratory tests.

p15

L202 - It is the participants who provide consent to participant in the study, not
the researchers. This presents a significant ethics issue for this study.

p18-19

Why was the change in anthropometric and immmunological parameters within group
important to analyse statistically? Given the objectives of the study, it is the
difference between the intervention and control groups that is relevant. The results
section is crowded with results that may not be important. In addition, the change
from baseline at the end of each month may be more useful than the change between
each month. Consider revising the results section and highlighting only the
difference in change from baseline between the intervention and control group. One
figure and one table, with legends, and focusing on the key results will further
enhance the section.

p24

L334 - There is no data on which the discussion on 'good adherence' is based. The
strategy the authors describe on p13 L161 was to monitor adherence, rather than
enhance or optimise it. No results are presented on the level of adherence. In fact
a large portion of the discussion refers to perceptions from participants and other
operational and implementation aspects that are core to the study and not presented
in the results and are appearing for the first time as discussion.

p21

L407 - From the power calculation, the study was adequately powered, perhaps a more
diverse population of PLWH rather than a larger sample size could be recommended?
The recommendations should be based on the effect on MO on CD4 and not extend to
issues of food security and nutrition which have not been demonstrated by the
current study.

Several challenges and limitations are cited. To what extend were participants
reluctant to keep clinic appointments? What was the effect on study outcomes and how
was that accounted for in the analysis?

The write up also has some grammatical errors and could benefit from revision by an
English editor. Some comments that may be helpful

Page 1

L1 - 'oleifera' is always written in lower case and itilised. Correct throughout the
manuscript.

L2 - Be consistent with the term used for study population. Ideally, use people first
language such as 'people living with HIV (PLWH)' rather than HIV patients or HIV
infected patients.

Page 9

L46 - Change accountable to accounts

L48 - revise grammar

L71 - 'leaf' powder

L72 - Sub-Saharan

6. PLOS authors have the option to publish the peer
review history of their article (what does this mean?). If published, this will
include your full peer review and any attached files.

If you choose “no”, your identity will remain anonymous but your review may still be
made public.

**Do you want your identity to be public for this peer review?** For
information about this choice, including consent withdrawal, please see our
Privacy Policy.

Reviewer #1: No

Reviewer #2: No

Reviewer #3: No

---

## [Author Response · Author response to Decision Letter 0]

4 May 2021

All comments have been responded to and attached as 'Answers to reviewers
comments'.

to Reviewers comments_ PLOS ONE.docx
---

## [Decision Letter · Decision Letter 1]

24 May 2021

PONE-D-20-30776R1

A double-blind randomized control trial to examine the effect of Moringa Oleifera
leaf powder supplementation on the anthropometric and immune status of adult HIV
patients on antiretroviral therapy in a resource- limited setting

PLOS ONE

Dear Dr. Gambo,

Thank you for submitting your manuscript to PLOS ONE. After careful consideration, we
feel that it has merit but does not fully meet PLOS ONE’s publication criteria after
your first revision. Therefore, we invite you to submit a further revised version of
the manuscript that addresses the outstanding statistical issues raised during the
current review process.

Please submit your revised manuscript by Jul 08 2021 11:59PM. If you will need more
time than this to complete your revisions, please reply to this message or contact
the journal office at plosone@plos.org. When
you're ready to submit your revision, log on to https://www.editorialmanager.com/pone/ and select the 'Submissions
Needing Revision' folder to locate your manuscript file.

If you would like to make changes to your financial disclosure, please include your
updated statement in your cover letter. Guidelines for resubmitting your figure
files are available below the reviewer comments at the end of this letter.

We look forward to receiving your revised manuscript.

Kind regards,

Tsitsi Grace Monera-Penduka, PhD, MSc, MPhil, BPharm Hons

Academic Editor

PLOS ONE

Journal Requirements:

Additional Editor Comments (if provided):

The authors have addressed the grammatical and other issue raise satisfactorily save
the statistical issues. They should revisit additional comments from the current
review and address them.

Reviewers' comments:

Reviewer's Responses to Questions

**Comments to the Author**

1. If the authors have adequately addressed your comments raised in a previous round
of review and you feel that this manuscript is now acceptable for publication, you
may indicate that here to bypass the “Comments to the Author” section, enter your
conflict of interest statement in the “Confidential to Editor” section, and submit
your "Accept" recommendation.

Reviewer #1: (No Response)

2. Is the manuscript technically sound, and do the data
support the conclusions?

Reviewer #1: Partly

3. Has the statistical analysis been performed
appropriately and rigorously? 

Reviewer #1: I Don't Know

4. Have the authors made all data underlying the
findings in their manuscript fully available?

Reviewer #1: Yes

5. Is the manuscript presented in an intelligible
fashion and written in standard English?

Reviewer #1: Yes

6. Review Comments to the Author

Reviewer #1: The authors appear to have at least partially implemented some changes
to the statistical analysis. However, it is not possible to tell whether the
analysis was performed correctly due to the lack of detail in the description of the
statistical analysis and in the results shown. From what can be seen, it appears
that the analysis was not specified correctly; or, if it was specified correctly, it
was not reported correctly.

Also, the authors should put some energy into evaluating whether the different types
of analysis performed produce similar results. My suggestion to keep the planned
analysis was because as a rule all pre-planned analyses should be presented unless
completely inappropriate. However, that means that there will be potentially
conflicting results, perhaps due to the effects of statistical correction, perhaps
due to increased or decreased power, or perhaps due to different assumptions.

In fact, this might result in relegating some part of the current work into a
supplementary appendix on the grounds that it is less germane.

The covariate analysis that is presented is also difficult to assess. The model
structure (as in the other case) is not clearly specified. If transformations were
used, they are not described. Note that "transformation" does not mean changing data
format from wide to long format. Instead, it means such things as logarithmic
transformation or other functional transformations, aggregation, or recoding. Simple
manipulation of data shape is irrelevant to the manuscript because it is only a
technical requirement of the data analysis software.

As initially mentioned in the first review, all linear model analysis must be
subjected to goodness of fit evaluation before interpretation. This is described in
brief in the first review and does not appear to have been performed.

My current recommendation is to obtain the services of a consulting statistician to
work through the data analysis and review comments and to put everything onto a
solid and consistent footing.

7. PLOS authors have the option to publish the peer
review history of their article (what does this mean?). If published, this will
include your full peer review and any attached files.

If you choose “no”, your identity will remain anonymous but your review may still be
made public.

**Do you want your identity to be public for this peer review?** For
information about this choice, including consent withdrawal, please see our
Privacy Policy.

Reviewer #1: No

---

## [Editor Report · Decision Letter 2]

15 Dec 2021

A double-blind, randomized controlled trial to examine the effect of Moringa oleifera
leaf powder supplementation on the immune status and anthropometric parameters of
adult HIV patients on antiretroviral therapy in a resource-limited setting.

PONE-D-20-30776R2

Dear Dr. Gambo,

We’re pleased to inform you that your manuscript has been judged scientifically
suitable for publication and will be formally accepted for publication once it meets
all outstanding technical requirements.

Kind regards,

Tsitsi G. Monera-Penduka

Guest Editor

PLOS ONE
---

## [Editor Report · Acceptance letter]

17 Dec 2021

PONE-D-20-30776R2 

A double-blind, randomized controlled trial to examine the effect of *Moringa
oleifera* leaf powder supplementation on the immune status and
anthropometric parameters of adult HIV patients on antiretroviral therapy in a
resource-limited setting 

Dear Dr. Gambo:

I'm pleased to inform you that your manuscript has been deemed suitable for
publication in PLOS ONE. Congratulations! Your manuscript is now with our production
department. 

Kind regards, 

on behalf of

Dr. Tsitsi G. Monera-Penduka 

Guest Editor

PLOS ONE